# Accuracy of C-reactive protein, procalcitonin, serum amyloid A and neopterin for low-dose CT-scan confirmed pneumonia in elderly patients: A prospective cohort study

Virginie Prendki[1,2], Astrid Malézieux-Picard[1]*, Leire Azurmendi[3], Jean-Charles Sanchez[2,3], Nicolas Vuilleumier[2,3,4], Sebastian Carballo[2,5], Xavier Roux[1,6], Jean-Luc Reny[2,5], Dina Zekry[1,2], Jérôme Stirnemann[2,5‡], Nicolas Garin[2,5,7‡], on behalf of the PneumOldCT study group[¶]

1 Department of Rehabilitation and Geriatrics, Division of Internal Medicine for the Aged, Geneva University Hospitals, Thônex, Switzerland, 2 Medical Faculty, Geneva, Switzerland, 3 Department of Internal Medicine Specialties, Medical Faculty, Geneva University Hospitals, Geneva, Switzerland, 4 Diagnostic Department, Division of Laboratory Medicine, Geneva University Hospitals, Geneva, Switzerland, 5 Department of Internal Medicine, Division of General Internal Medicine, Geneva University Hospitals, Geneva, Switzerland, 6 Department of Intensive Care Unit, Geneva University Hospitals, Geneva, Switzerland, 7 Department of General Internal Medicine, Division of General Internal Medicine, Riviera Chablais Hospitals, Rennaz, Switzerland

☯ These authors contributed equally to this work.
‡ JS and NG also contributed equally to this work.
¶ Membership of the PneumOldCT study group is listed in the Acknowledgments.
* virginie.prendki@hcuge.ch

## Abstract

### Objective

The diagnosis of pneumonia based on semiology and chest X-rays is frequently inaccurate, particularly in elderly patients. Older (C-reactive protein (CRP); procalcitonin (PCT)) or newer (Serum amyloid A (SAA); neopterin (NP)) biomarkers may increase the accuracy of pneumonia diagnosis, but data are scarce and conflicting. We assessed the accuracy of CRP, PCT, SAA, NP and the ratios CRP/NP and SAA/NP in a prospective observational cohort of elderly patients with suspected pneumonia.

### Methods

We included consecutive patients more than 65 years old, with at least one respiratory symptom and one symptom or laboratory finding suggestive of infection, and a working diagnosis of pneumonia. Low-dose CT scan and comprehensive microbiological testing were done in all patients. The index tests, CRP, PCT, SAA and NP, were obtained within 24 hours. The reference diagnosis was assessed *a posteriori* by a panel of experts considering all available data, including patients' outcome. We used area under the curve (AUROC) and Youden index to assess the accuracy and obtain optimal cut-off of the index tests.

**Data Availability Statement:** All relevant data are within the manuscript and its Supporting Information files.

**Funding:** The PneumOldCT study was supported by grants from the Geneva University Hospitals (HUG) (Research & Development Grant, Medical Directorate, HUG), the Department of Internal Medicine of the University Hospital and the Faculty of Medicine of Geneva and the Ligue Pulmonaire Genevoise, a non-profit association involved in the care of patients with respiratory diseases.

**Competing interests:** The authors have declared that no competing interests exist.

## Results

200 patients (median age 84 years) were included; 133 (67%) had pneumonia. AUROCs for the diagnosis of pneumonia was 0.64 (95% CI: 0.56–0.72) for CRP; 0.59 (95% CI: 0.51–0.68) for PCT; 0.60 (95% CI: 0.52–0.69) for SAA; 0.41 (95% CI: 0.32–0.49) for NP; 0.63 (95% CI: 0.55–0.71) for CRP/NP; and 0.61 (95% CI: 0.53–0.70) for SAA/NP. No cut-off resulted in satisfactory sensitivity or specificity.

## Conclusions

Accuracy of traditional (CRP, PCT) and newly proposed biomarkers (SAA, NP) and ratios of CRP/NP and SAA/NP was too low to help diagnosing pneumonia in the elderly. CRP had the highest AUROC.

## Clinical Trial Registration

NCT 02467092

## Introduction

Pneumonia is the leading cause of death from infectious disease in elderly patients, and is frequently suspected in the emergency department. In addition of suggestive symptoms and signs, the diagnosis requires the presence of a new infiltrate on radiologic imaging, usually chest X-ray (CXR). However, in the elderly, symptoms and signs are often atypical, not specific, or lacking, and other frequent diseases can mimic pneumonia (eg. heart failure, chronic obstructive pulmonary disease). Finally, CXR is often inconclusive. These drawbacks may cause under- or overtreatment [1–3]. Prospective diagnostic studies have found a better accuracy of CT scan compared with CXR in adult and elderly patients suspected of pneumonia [4, 5]. CT-scan confirmed pneumonia is currently the best reference diagnosis for pneumonia [6].

Serum biomarkers are potential valuable diagnostic tools, and are more convenient to use than CT-scan. C-reactive protein (CRP) and procalcitonin (PCT) have been widely tested for the diagnosis of pneumonia in adults, but few studies have included elderly patients [7, 8]. In recent studies, serum Amyloid A (SAA) and neopterin (NP) were predictive of stroke-associated infections, especially pneumonia [9, 10]. The ratio CRP/NP has been proposed to discriminate an exacerbation of chronic obstructive pulmonary disease (COPD) from pneumonia [11].

We hypothesized that biomarkers would improve the clinical and radiological diagnosis of pneumonia, and assessed the accuracy of CRP, PCT, SAA, NP and the ratios CRP/NP and SAA/NP in the PneumOldCT study, a cohort of elderly patients with CT-scan confirmed pneumonia. We compared accuracy of the biomarkers with accuracy of the usual diagnostic process and reported their optimal cut-off value.

## Materials and methods

### Setting and participants

This is a diagnostic study nested in a prospective observational cohort. Consecutive patients older than 65 years and hospitalized with suspected pneumonia in Geneva University Hospitals, a 1800-bed tertiary-care hospital, were eligible [5]. Patients had at least one respiratory

symptom or sign and one symptom, sign, or laboratory finding suggestive of acute infection, with a working diagnosis of pneumonia warranting antibiotic treatment [5]. The choice and duration of antibiotic treatment were at the discretion of the treating physician. Patients diagnosed with pneumonia during the previous 6 months, or treated with antibiotics for more than 48 hours before inclusion, were excluded. All patients had CXR and low-dose CT-scan (LDCT) without injection of contrast medium performed within 24 hours after inclusion. The study was approved by Geneva's Institutional Review Board (CER-14-250) and registered at clinicaltrials.gov (NCT02467192). Informed consent was obtained from all patients or next of kin.

### Recorded data

Demographic data, comorbidities, vital signs, clinical findings, severity scores of pneumonia and the results of laboratory tests (including CRP and PCT) were recorded at admission. Comprehensive microbiological testing was performed in all patients. It consisted of naso-pharyngeal swabs (NPS) for detection of common viral pathogens by polymerase chain reaction (PCR); blood and sputum cultures; and testing for *Streptococcus pneumoniae* and *Legionella pneumophila* antigenuria. Only high quality sputum samples were sent to culture.

### Index tests

Blood samples for CRP, PCT, SAA and NP were obtained within 24 hours after admission. CRP and PCT measurement were performed in routine care and SAA and NP were measured retrospectively. Plasma CRP concentrations were measured via immunoturbidimetry (Roche/ Hitachi Cobas c702 systems) and PCT using a rapid assay with a sensitivity of 0.06 μg/L (Kryptor, Brahms, Hennigsdorf). Levels of SAA in plasma were determined using an electrochemiluminescence detection system using multi-array technology (SECTOR Imager 2400, MSD, Gaithersburg) [12]. Determination of NP was performed using a competitive enzyme-linked immunosorbent assay (ELISA) (ELItest® Neopterin-Screening, Brahms).

### Usual diagnosis and reference diagnosis

The physician in charge of the patient was asked to rate the probability of pneumonia before the performance of LDCT on a three-level Likert scale (low, intermediate and high), based on the results of routine blood tests and CXR. The diagnosis of pneumonia was considered positive (respectively "negative") if the rated probability of pneumonia was "intermediate" or "high" (respectively "low"). This diagnosis was used to assess the accuracy of the usual diagnostic process.

The reference diagnosis was assessed *a posteriori* by senior physicians (experts) experienced in the diagnosis and management of pneumonia, including radiologists specialized in thoracic imaging. The experts had access to all clinical, biological (including CRP and PCT, but not SAA and NP), and microbiological data. They were aware of patients' evolution and final outcomes, and had access to all CXR and LDCT imagings. They rated the probability of pneumonia according to the same Likert scale as the emergency physician. Discordant cases were reviewed using a Delphi method until consensus was reached. The reference diagnosis was considered positive (respectively "negative") if the panel of experts rated the probability of pneumonia "intermediate" or "high" (respectively "low") on the Likert scale. In a sensitivity analysis, only patients with a high probability of disease were considered positive.

## Data analysis

Sample size is based on the power calculation of the original study [5]. We used frequencies, percentage, and median with interquartile range for descriptive purposes. Variables were compared between patients with and without pneumonia in univariate analysis using the Mann-Whitney-Wilcoxon test or the Student's test for continuous variables, and Fisher's exact test or Chi-square test for categorical variables, as appropriate. We computed sensitivity, specificity, positive and negative predictive values, we constructed Receiver Operating Characteristic curves for each biomarker and obtained the AUROC with 95% CI. The best cut-off value of biomarkers was determined with the Youden index. The AUROCs were compared using De Long test.

To assess if use of any tested biomarker adds information on top of routinely collected clinical variables, we built a clinical score predicting the presence of pneumonia, using clinical symptoms and signs present at admission and associated with the diagnosis in univariate analysis (p< 0.20) and obtained the AUROC of the score with 95% CI. We then added separately each tested biomarker, dichotomized at the best predicted cut-off, to the clinical score, and computed AUROC of the new score.

All p values are two-tailed and considered significant for p<0.05. Data were analyzed using the R statistical software package, version 3.1.1 (R Core Team (2020). R: A language and environment for statistical computing. R Foundation for Statistical Computing, Vienna, Austria. URL https://www.R-project.org/), and SPSS (IBM SPSS Statistics for Windows, Version 25.0. Armonk, NY: IBM Corp).

## Results

### Patients' characteristics

Of 899 patients screened, 200 (median age 84 years, IQR: 78.6–90.2) were included. CRP was available in all patients and PCT in 185. SAA and NP were measured *a posteriori* in 192 patients. Sex ratio was approximately 1. Forty-seven percent of the patients were 85 years or older. Twenty-one patients (10.5%) lived in a nursing home. Active smokers had more frequently pneumonia. (p = 0.08). Main comorbidities were cardiovascular diseases (n = 103, 51.5%), cognitive impairment (n = 66, 34.5%) and chronic renal failure (n = 60, 30.0%). Cough (n = 170, 85%), dyspnea (n = 145, 72.5%) and crackles (n = 171, 85.5%) were the most frequent symptoms and signs. The median CURB65 score was 2, and 30-day mortality was 7%. Expert panel classified 99 patients (49.5%) as having high, 34 (17.0%) intermediate, and 67 (33.5%) low probability of disease. Hence, expert panel classified pneumonia as present in 133 patients and absent in 67. In patients with pneumonia, a pathogen was identified in 55 (41.4%). The main characteristics of these two groups are reported in Table 1.

### Index tests results

The median levels of all four biomarkers differed significantly between patients with and without pneumonia (Fig 1). The median values of CRP were 62.7 mg.L$^{-1}$ [38.5–107.5] and 100.7 mg.L$^{-1}$ [59.0–205.2] (p = 0.001), and of PCT 0.2 µg.L$^{-1}$ [0.1–0.7] and 0.4 µg.L$^{-1}$ [0.1–1.9] (p<0.05) in patients with and without pneumonia, respectively. The corresponding values of SAA were 262.0 µg.L$^{-1}$ [231.8–278.0] and 267.5 µg.L$^{-1}$ [252.0–287.8] (p<0.05), and of NP 9.3 nmol.L$^{-1}$ [5.9–14.0] and 6.6 nmol.L$^{-1}$ [4.5–12.6] in patients with and without pneumonia.

AUROC for pneumonia diagnosis was 0.64 (95% CI: 0.56–0.72) for CRP; 0.59 (95% CI: 0.51–0.68) for PCT; 0.60 (95% CI: 0.52–0.69) for SAA; 0.41 (95% CI: 0.32–0.49) for NP; 0.63 (95% CI: 0.55–0.71) for CRP/NP and 0.61 (95% CI: 0.53–0.70) for SAA/NP (Fig 2). The

**Table 1. Baseline characteristics of the 200 patients included.**

| Characteristics | | No. (%) or Median [IQR] | |
|---|---|---|---|
| | All patients (n = 200 unless stated) | Pneumonia excluded (n = 67) | Pneumonia confirmed (n = 133) |
| *Demographics* | | | |
| Age (years) | 84 [79–90] | 86 [80–92] | 83 [78–89] |
| Female gender | 98 (49.0) | 38 (56.7) | 60 (45.1) |
| Active smoker, n(%) (n = 199) | 34 (17.0) | 7 (10.4) | 27 (20.3) |
| Living place | | | |
| Home | 172 (86.0) | 55 (82.1) | 117 (88.0) |
| Nursing home | 21 (10.5) | 10 (15.0) | 11 (8.3) |
| Other | 7 (3.5) | 2 (2.9) | 5 (3.7) |
| *Comorbidities* | | | |
| Cardiovascular disease | 103 (51.5) | 37 (55.2) | 66 (49.6) |
| Chronic obstructive pulmonary disease (n = 197) | 35 (17.8) | 9 (13.4) | 26 (20.0) |
| Chronic renal disease | 60 (30.0) | 24 (35.8) | 36 (27.1) |
| Cerebrovascular disease | 31 (15.5) | 11 (16.4) | 20 (15.0) |
| Immunosuppressive therapy (n = 199) | 15 (7.5) | 3 (4.5) | 12 (9.0) |
| Diabetes mellitus | 45 (22.5) | 13 (19.4) | 32 (24.1) |
| Chronic liver disease | 11 (5.5) | 3 (4.5) | 8 (6.0) |
| Active cancer (n = 196) | 17 (8.7) | 3 (4.6) | 14 (10.7) |
| Swallowing disorders (n = 177) | 28 (15.8) | 9 (14.1) | 19 (16.8) |
| Poor oral health (n = 175) | 38 (21.7) | 11 (19.0) | 27 (23.1) |
| Cognitive impairment (n = 193) | 66 (34.5) | 27 (42.2) | 39 (30.7) |
| *Scores* | | | |
| Charlson comorbidity index (n = 165) | | | |
| Mean | 3 (1–10) | 3 (1–10) | 3 (1–9) |
| Score = 1, n(%) | 39 (23.6) | 14 (25.5) | 25 (22.7) |
| Score = 2, n(%) | 44 (26.7) | 16 (29.1) | 28 (25.5) |
| Score >2, n(%) | 82 (49.7) | 30 (45.4) | 57 (51.8) |
| Mini mental state examination n = 162) | 24 (19–27) | 22 (15–27) | 24 (19–27) |
| Mini nutritional assessment (n = 178) | 8 (6–11) | 8 (6–10) | 9 (6–11) |
| *Symptoms and signs at admission* | | | |
| Confusion | 92 (46.0) | 32 (47.8) | 60 (45.1) |
| Falls | 71 (35.5) | 29 (43.3) | 42 (31.6) |
| Respiratory rate >20/min | 158 (79.4) | 49 (73.1) | 109 (82.6) |
| Fever (temperature >37.8 ˚C) | 103 (51.5) | 29 (43.3) | 84 (55.6) |
| Cough | 170 (85.0) | 50 (74.6) | 120 (90.2) |
| Sputum production | 74 (37.0) | 25 (37.3) | 49 (36.8) |
| Chest pain | 35 (17.5) | 9 (13.4) | 26 (19.5) |
| Dyspnea | 145 (72.5) | 50 (74.6) | 95 (71.4) |
| Crackles | 171 (85.5) | 57 (85.1) | 114 (85.7) |
| Oxygen saturation <90% | 102 (51.0) | 30 (44.8) | 72 (54.1) |
| Pulse rate >125/min | 13 (6.5) | 6 (8.9) | 7 (5.3) |
| SBP <90 mmHg or DBP < ...60 mmHg | 34 (17.0) | 10 (14.9) | 24 (18.0) |
| *Laboratory findings* | | | |
| Leukocytes ($10^3$ per $mm^3$) | 11.0 [8.2–14.0] | 10.7 [7.9–13.1] | 11.3 [8.6–14.7] |
| Urea ($mg.L^{-1}$) | 7.9 [6.0–11.9] | 8.3 [6.2–12.8] | 7.7 [5.7–10.8] |
| NT-proBNP ($ng.L^{-1}$) | 1836.5 [666.8–3800.8] | 1884.0 [649.5–3550.0] | 1826.0 [685.5–3860.5] |
| Prealbumin ($g.L^{-1}$) | 122.0 [95.0–162.0] | 131.0 [99.0–167.0] | 118.5 [86.7–157.2] |

(*Continued*)

**Table 1.** (Continued)

| Characteristics | | No. (%) or Median [IQR] | |
|---|---|---|---|
| | **All patients (n = 200 unless stated)** | *Pneumonia excluded (n = 67)* | *Pneumonia confirmed (n = 133)* |
| Albumin (g.L$^{-1}$) | 35.0 [32.0–38.0] | 35.0 [32.0–37.0] | 35.0 [31.0–38.0] |
| C-reactive protein (mg.L$^{-1}$) | 84.0 [45.8–159.7] | 62.7 [38.5–107.5] | 100.7 [59.0–205.2] |
| Procalcitonin (µg.L$^{-1}$) | 0.3 [0.1–1.3] | 0.2 [0.1–0.7] | 0.4 [0.1–1.9] |
| Serum amyloid A (µg.L$^{-1}$) | 265.0 [247.8–285.2] | 262.0 [231.8–278.0] | 267.5 [252.0–287.8] |
| Neopterin (nmol.L$^{-1}$) | 7.6 [4.7–13.4] | 9.3 [5.9–14.0] | 6.6 [4.5–12.6] |
| *Pathogen identified, n(%)* | | | |
| Bacterial | 22 (11.0) | 4 (6.0) | 18 (32.7) |
| Viral | 62 (31.0) | 25 (37.3) | 37 (67.3) |
| None | 116 (58.0) | 38 (56.7) | 78 (58.6) |
| *Vaccination status, n(%)* | | | |
| Influenza vaccination (n = 182) | 103 (56.6) | 36 (63.2) | 67 (53.6) |
| Pneumococcal vaccination (n = 177) | 7 (3.9) | 2 (3.6) | 5 (4.1) |
| *Disease severity, n(%)* | | | |
| PSI score | 102 [87–121] | 104 [87–121] | 98 [86–121] |
| CURB 65 | 2 [2–3] | 2 [2–3] | 2 [2–3] |
| CURB 65 >2 | 89 (44.5) | 28 (41.8) | 61 (45.9) |
| *Outcome* | | | |
| 30-day mortality | 14 (7.0) | 4 (6.0) | 10 (7.5) |
| 90-day mortality (n = 198) | 29 (14.6) | 10 (14.9) | 19 (14.3) |

Laboratory values and vital signs were obtained at hospital admission.

Definitions: Immunosuppressive therapy: prednisone for more than two weeks; or receipt of other immunosuppressive drugs. Cognitive impairment was diagnosed after geriatrician evaluation (at least CDR 1 dementia). Swallowing disorders: observed during the hospitalization

Oral health rated as good, medium or poor

Abbreviations: CURB65 is a pneumonia severity score based on confusion, respiratory rate, blood pressure, and age 65 or older. DBP: diastolic blood pressure SBP: systolic blood pressure

AUROC of CRP did not differ significantly from the AUROC of any other biomarker by de Long test (results not shown). The AUROC of the usual diagnostic process was 0.55 (95% CI 0.46–0.64). The AUROC of all four biomarkers using the alternative reference diagnosis definition (considering only the patients with a high probability of disease as positive) were similar and are not reported.

The cut-off values of CRP, PCT, SAA, NP, CRP/NP, and SAA/NP calculated with Youden method, were set at 109.4 mg.L$^{-1}$ (sensitivity 50% and specificity 76%), 1.1 µg.L$^{-1}$ (37% and 83%), 282.0 µg.L$^{-1}$ (39% and 81%), 16.4 nmol.L$^{-1}$ (19% and 84%), 15.24 (45% and 81%) and 36.41 (46% and 77%), respectively (Table 2). No cut-off resulted in sensitivity or specificity values likely to be useful in clinical practice.

Cough, tachypnea, fever, and falls (as presenting signs or symptoms) were associated with pneumonia with a p value < 0.20. We dismissed falls as this is not widely described as a predictor of pneumonia, and built a clinical score by adding one point for the presence of each of cough, tachypnea and fever. AUROC of the clinical score and AUROCs of scores obtained by adding one point for each dichotomized biomarker to the clinical score are compared in Table 3.

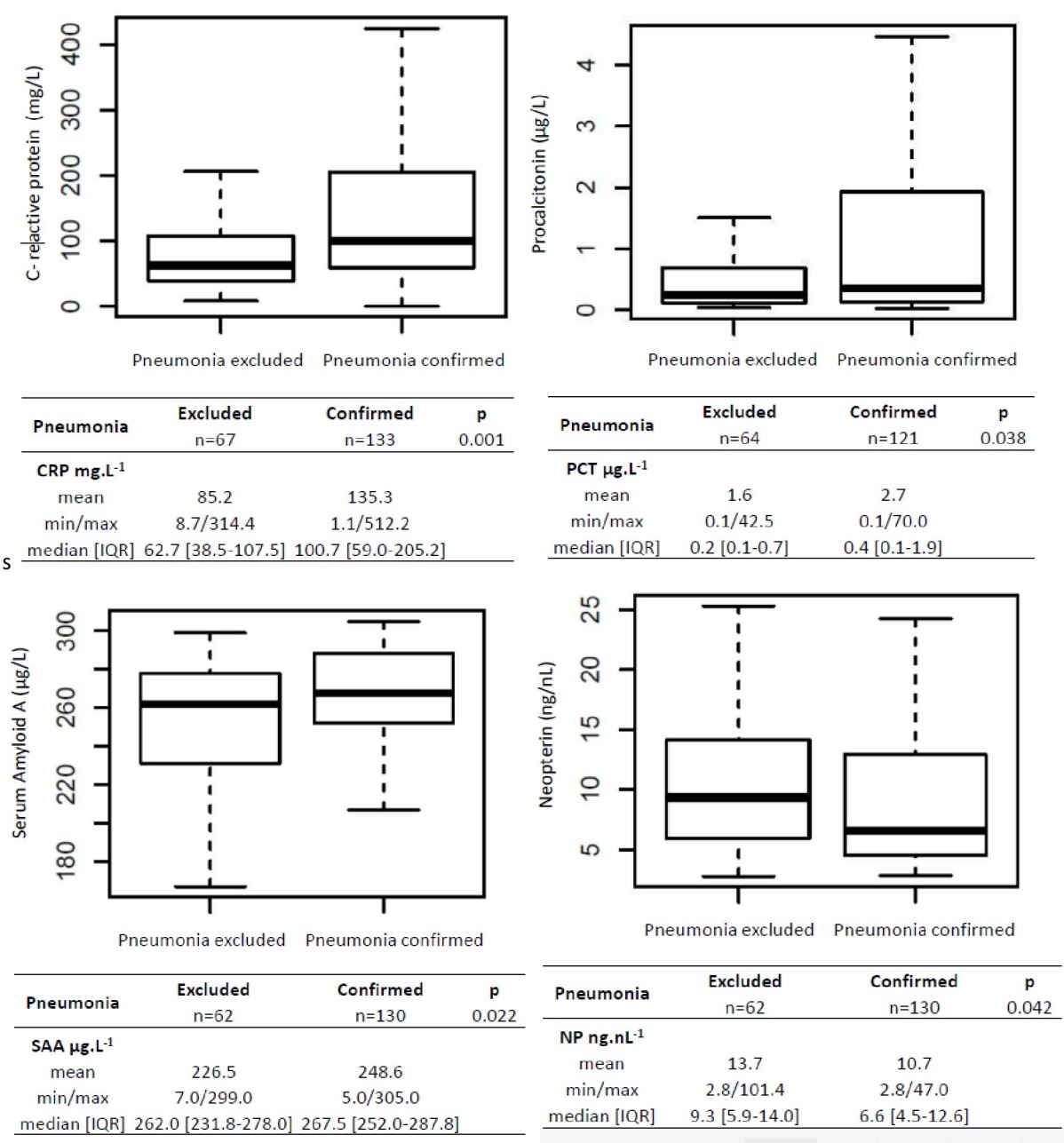

| Pneumonia | Excluded n=67 | Confirmed n=133 | p 0.001 |
|---|---|---|---|
| CRP mg.L⁻¹ | | | |
| mean | 85.2 | 135.3 | |
| min/max | 8.7/314.4 | 1.1/512.2 | |
| median [IQR] | 62.7 [38.5-107.5] | 100.7 [59.0-205.2] | |

| Pneumonia | Excluded n=64 | Confirmed n=121 | p 0.038 |
|---|---|---|---|
| PCT µg.L⁻¹ | | | |
| mean | 1.6 | 2.7 | |
| min/max | 0.1/42.5 | 0.1/70.0 | |
| median [IQR] | 0.2 [0.1-0.7] | 0.4 [0.1-1.9] | |

| Pneumonia | Excluded n=62 | Confirmed n=130 | p 0.022 |
|---|---|---|---|
| SAA µg.L⁻¹ | | | |
| mean | 226.5 | 248.6 | |
| min/max | 7.0/299.0 | 5.0/305.0 | |
| median [IQR] | 262.0 [231.8-278.0] | 267.5 [252.0-287.8] | |

| Pneumonia | Excluded n=62 | Confirmed n=130 | p 0.042 |
|---|---|---|---|
| NP ng.nL⁻¹ | | | |
| mean | 13.7 | 10.7 | |
| min/max | 2.8/101.4 | 2.8/47.0 | |
| median [IQR] | 9.3 [5.9-14.0] | 6.6 [4.5-12.6] | |

**Fig 1. C-reactive protein (CRP), procalcitonin (PCT), Serum Amyloid A (SAA) and neopterin (NP) boxplot in patients with and without pneumonia.**

## Discussion

This study assesses the diagnostic value of serum biomarkers in a cohort of elderly patients suspected of pneumonia. The accuracy of traditional infection biomarkers (CRP and PCT) was low, and newly proposed biomarkers (SAA, NP) and ratios of CRP/NP and SAA/NP were not significantly better. Nevertheless, most biomarkers had a slightly better accuracy than the physician diagnosis following the usual diagnostic process, i.e. using clinical symptoms and signs and CXR (AUROC 0.55). The best AUROC was 0.64 for CRP.

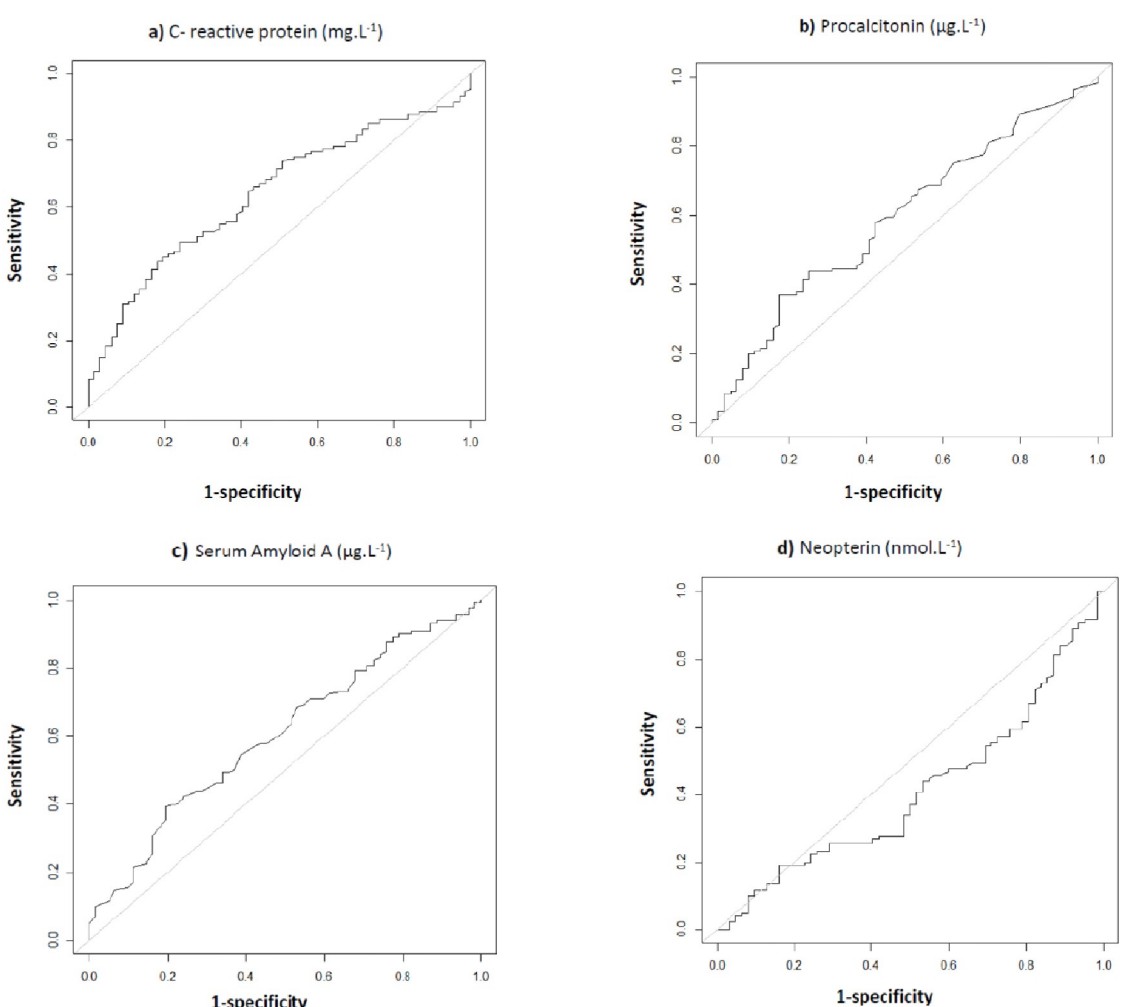

**Fig 2. ROC curves for the diagnosis of pneumonia.** a) C-reactive protein Area under the curve [95% CI] = 0.64 [0.56–0.72]. Optimal cut-off point at 109.4 mg.L$^{-1}$. b) Procalcitonin Area under the curve [95% CI] = 0.59 [0.51–0.68]. Optimal cut-off point at 1.06 µg.L$^{-1}$. c) Neopterin Area under the curve [95% CI] = 0.41 [0.32–0.49]. Optimal cut-off point at 16.4 nmol.L$^{-1}$. d) Serum amyloid A Area under the curve [95% CI] = 0.60 [0.52–0.69]. Optimal cut-off point at 15.3 µg.L$^{-1}$.

**Table 2. Sensitivity, specificity, positive and negative predictive values according to C-reactive protein, procalcitonin, serum amyloid A, neopterin, C-reactive protein/neopterin and serum amyloid A/neopterin at the best cut-off values (computed with Youden index).**

|  | CRP (mg.L$^{-1}$) | PCT (µg.L$^{-1}$) | SAA (µg.L$^{-1}$) | NP (nmol.L$^{-1}$) | CRP/NP | SAA/NP |
|---|---|---|---|---|---|---|
| **AUROC (95%CI)** | 0.64 (0.56–0.72) | 0.59 (0.51–0.68) | 0.60 (0.52–0.69) | 0.41 (0.32–0.49) | 0.63 (0.55–0.71) | 0.61 (0.53–0.70) |
| **Cut-off** | 109.4 | 1.1 | 282.0 | 16.4 | 15.24 | 36.41 |
| **Sensitivity** | 50% | 37% | 39% | 19% | 45% | 46% |
| **Specificity** | 76% | 83% | 81% | 84% | 81% | 77% |
| **PPV** | 80% | 80% | 81% | 71% | 83% | 81% |
| **NPV** | 43% | 41% | 39% | 33% | 41% | 41% |

Abbreviations: AUROC (Area Under the Receiver Operating Curve), CRP for C-reactive protein, PCT for procalcitonin, SAA for serum amyloid A, NP for neopterin, PPV for positive predictive values, NPV for negative predictive values

**Table 3. AUROCs of a clinical score with and without biomarkers for the prediction of pneumonia.**

| | AUROC clinical score (95% CI) | AUROC clinical score plus biomarker (95% CI) |
|---|---|---|
| CRP (cut-off: 110 mg L$^{-1}$) | 0.63 (0.55–0.71) | 0.68 (0.60–0.76) |
| CRP / neopterin (cut-off 15) | 0.63 (0.55–0.71) | 0.67 (0.59–0.75) |
| SAA / neopterin (cut-off 37.4) | 0.63 (0.55–0.71) | 0.66 (0.58–0.74) |
| SAA (cut-off 282.0 μg. L$^{-1}$) | 0.63 (0.55–0.71) | 0.65 (0.56–0.73) |
| PCT (cut-off 1.1 μg. L$^{-1}$) | 0.63 (0.55–0.71) | 0.65 (0.57–0.73) |

Adding a biomarker dichotomized at the best predicted cut-off to a score based on clinical variables (cough, fever and tachypnea) resulted in a higher accuracy. Adding CRP increased AUROC from 0.63 to 0.68 for example. However, the AUROC remains disappointingly low.

Pneumonia is a highly heterogeneous disease, which may explain why the diagnostic approach based on semiology and CXR is frequently inaccurate, particularly in the elderly [13–15]. Inclusion of biomarkers in the diagnostic pathway has been advocated to enhance its accuracy. In a large study conducted in the primary care setting by van Vugt et al., CRP at a cut-off of 30 mg.L$^{-1}$ modestly improved diagnostic classification, a finding comparable to our results [16]. Minaard et al. confirmed in an individual patient data meta-analysis that adding CRP to clinical prediction models improved reclassification of pneumonia in 15% of the patients [17]. Of note, the reference diagnosis in these studies used CXR as the diagnostic imaging modality, which has been shown to convey a substantial risk of misclassification [1, 4, 5].

A reference diagnosis incorporating the results of CT-scan in all patients was used to assess the accuracy of CRP and PCT in a prospective study of 200 patients (median age 64 years) presenting at the emergency room with suspected pneumonia [18]. AUROCs of CRP and PCT for the diagnosis of CAP were 0.79 and 0.66, respectively. In our study using a similar reference diagnosis, we found lower AUROCs for CRP (0.64) and PCT (0.59). This discrepancy could stem from different included populations, our patients being 20 years older. Few clinical trials have assessed biomarkers in an elderly population, because including such patients is challenging [19]. Stucker et al. showed that higher CRP, but not PCT, was associated with an acute infection in a prospective cohort of patients over 75 years admitted to a geriatric hospital [8]. In a retrospective study including elderly patients (median age 81 years) hospitalized for an acute respiratory infection, AUROCs for the diagnosis of pneumonia were 0.76 for CRP and 0.54 for PCT [20].

The best cut-off for CRP in our study was 109.4 mg.L$^{-1}$ In comparison, the optimal cut-off for CRP was 50 mg.L$^{-1}$ in the aforementioned study by Le Bel et al. (median age: 64 years), and 30 mg.L$^{-1}$ in the study by van Vugt et al. (median age: 50 years) [16], but 61 mg. L$^{-1}$ in a cohort of multimorbid elderly patients hospitalized for respiratory symptoms [20]. This variation in the optimal threshold is probably due to a higher background CRP in elderly patients with multiple comorbidities [21].

In our study as in most previous reports, CRP outperformed PCT for the diagnosis of pneumonia. This is not surprising, as the proposed role of PCT in respiratory infections is not to diagnose pneumonia but to identify patients that can be managed safely without antibiotics [22]. Procalcitonin in patients admitted to an acute geriatric ward did not discriminate patients with infection [8], and had limited clinical usefulness to diagnose invasive bacterial infections [7]. Chronic low-grade inflammation and lower eGFR might result in elevated baseline levels of PCT in elderly patients [23].

SAA is an acute-phase protein mostly synthesized by the liver, with a significantly shorter half-life than CRP [24]. In the pediatric setting, SAA predicted the presence of ventilator-acquired pneumonia with a sensitivity of 100% and a specificity of 93% [25]. Azurmendi et al. showed that SAA was elevated in patients at risk to develop post-stroke infections [26]. Nevertheless it had poor accuracy in our study, with an AUROC of 0.60, as well as SAA/NP with an AUROC 0.61.

NP is a marker of cell-mediated immunity, produced by monocytes and macrophages upon stimulation with interferon-gamma [27]. Pizzini et al. found that CRP/NP could discriminate pneumonia from acute exacerbation of COPD [11]. In our study, CRP/NP was not better than CRP alone.

Biomarkers levels vary according to immune status, the nature of the pathogen, the extension of the infection, and timing of the measurement relative to the beginning of the infection [28]. Smoking is also associated with alterations in the level of inflammatory markers. In our cohort, smokers were overrepresented in patients with pneumonia. [29] All these confounding factors may explain why the quest for a biomarker to diagnose pneumonia remains unsuccessful. In elderly patients, frequently present cardiovascular, respiratory, oncologic, and neurodegenerative diseases may further confound the relation between a biomarker and an acute infectious disease. Moreover, elderly patients frequently present with a chronic, low-grade inflammation of undetermined origin, called inflammaging [30]. To surpass these limitations, currently proposed strategies use simultaneous dosing of multiple viral- and bacterial-induced host proteins [31]. Other strategies combine biological, microbiological and radiological data into scores or decision rules [6, 32]. Another approach could be the sequential use of two biomarkers, the first with a high sensitivity and the second with a high specificity.

Our study has several strengths. It was conducted in a consecutive cohort of 200 elderly multimorbid patients representative of real life practice. We used a robust reference standard based on assessment of all data by a panel of experts and including thoracic CT scan and comprehensive microbiological testing in all patients. Our study has also limitations. First, the generalizability of the results is limited because it was conducted in a single hospital. Second, the expert panel was blinded to SAA and NP results, but not to CRP and PCT. This might have artificially inflated the accuracy of the latter. Third, we could not assess precisely the beginning of symptoms, information difficult to obtain in elderly patients with frequent cognitive impairment and delirium. Fourth, we could not compute the net reclassification improvement by each biomarker added to the usual diagnostic process.

In conclusion, the diagnosis of pneumonia in the elderly is often uncertain, and neither traditional nor newly proposed biomarkers had sufficient accuracy to be useful in this diagnosis. Further research should focus on scores or decision rules combining clinical, biological and radiological data. Simultaneous use of several biomarkers reflecting different aspects of the complex pathophysiology of pneumonia should also be tested. Finally, future studies should assess the net reclassification improvement by any new biomarker as compared to the usual diagnostic process.

## Supporting information

**S1 File.**
(XLSX)

## Acknowledgments

We thank the patients and their families for their participation in this study. We also thank all members of PneumOldCT study group, clinicians, radiology technicians, research nurses, the case managers who helped us enrol our participants as well as the Clinical Research Center of Hôpitaux Universitaires de Genève (HUG). We acknowledge the contribution of the ESCMID Study Group for Infections in the Elderly (ESGIE, www.escmid.org/esgie).

**Complete membership of the PneumOldCT study group**:

The leader of the group is Dr Virginie Prendki (principal investigator), virginie.prendki@h-cuge.ch.

Other members: (in alphabetical order):

T Agoritsas, S Carballo, P Darbellay Farhoumand, C Marti, JL Reny, S Rosset-Zufferey, Jacques Serratrice, V. Soulier, J Stirnemann (co-investigator): Division of Internal Medicine, Department of Internal Medicine Specialties, Geneva University Hospitals, Switzerland

C. Combescure: Clinical Research Center, Geneva University and Hospitals Geneva University, Switzerland

N Garin: Department of General Internal Medicine, Riviera Chablais Hospitals, Switzerland

F Herrmann, V Lachat, MP Meynet, X Roux, C Serratrice, Department of Rehabilitation and Geriatrics, Geneva University Hospitals, Switzerland

X Montet, M Scheffler, Department of Radiology, Geneva University Hospitals and University of Geneva, Switzerland

B Huttner, L Kaiser. Division of Infectious Diseases, Department of Internal Medicine Specialties, Medical Faculty, Geneva University and Hospitals Geneva University, Switzerland

JP Janssens, Division of Pulmonology, Department of Internal Medicine Specialties, Geneva University Hospitals and University of Geneva, Switzerland

**Sponsor:** HUG

## Author Contributions

**Conceptualization:** Virginie Prendki, Jérôme Stirnemann.

**Formal analysis:** Virginie Prendki, Astrid Malézieux-Picard, Leire Azurmendi, Jérôme Stirnemann, Nicolas Garin.

**Funding acquisition:** Virginie Prendki.

**Investigation:** Virginie Prendki, Astrid Malézieux-Picard, Leire Azurmendi, Jean-Charles Sanchez, Nicolas Vuilleumier.

**Methodology:** Virginie Prendki, Jean-Charles Sanchez, Nicolas Vuilleumier, Jérôme Stirnemann, Nicolas Garin.

**Project administration:** Virginie Prendki.

**Resources:** Leire Azurmendi, Jean-Charles Sanchez, Nicolas Vuilleumier.

**Software:** Leire Azurmendi, Nicolas Garin.

**Supervision:** Virginie Prendki, Jean-Charles Sanchez, Nicolas Garin.

**Validation:** Jean-Charles Sanchez, Nicolas Vuilleumier, Sebastian Carballo, Xavier Roux, Jean-Luc Reny, Jérôme Stirnemann, Nicolas Garin.

**Visualization:** Virginie Prendki, Sebastian Carballo, Xavier Roux, Jean-Luc Reny, Dina Zekry, Nicolas Garin.

**Writing – original draft:** Virginie Prendki, Astrid Malézieux-Picard.

**Writing – review & editing:** Virginie Prendki, Astrid Malézieux-Picard, Jean-Charles Sanchez, Nicolas Vuilleumier, Sebastian Carballo, Xavier Roux, Jean-Luc Reny, Dina Zekry, Jérôme Stirnemann, Nicolas Garin.

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
