## [Decision Letter · Decision Letter 0]

1 Jul 2020

PONE-D-20-15597

Accuracy of C-reactive protein, procalcitonin, serum amyloid A and neopterin for low-dose CT-scan confirmed pneumonia in elderly patients: a prospective cohort study

PLOS ONE

Dear Dr. Prendki,

Thank you for submitting your manuscript to PLOS ONE. After careful consideration, we feel that it has merit but does not fully meet PLOS ONE’s publication criteria as it currently stands. Therefore, we invite you to submit a revised version of the manuscript that addresses the points raised during the review process.

 Please find below the comments by the reviewers. In your revision, please provide point-by-point to their queries. We look forward to your revised manuscript.

We look forward to receiving your revised manuscript.

Kind regards,

Muhammad Adrish

Academic Editor

PLOS ONE

Journal Requirements:

2. In your Methods section, please provide additional details regarding the sample size calculation employed to justify the number of patients included in this study.

3. One of the noted authors is a group or consortium: PneumOldCT study group.

In addition to naming the author group, please list the individual authors and affiliations within this group in the acknowledgments section of your manuscript.

Please also indicate clearly a lead author for this group along with a contact email address.

Reviewers' comments:

Reviewer's Responses to Questions

**Comments to the Author**

1. Is the manuscript technically sound, and do the data support the conclusions?

Reviewer #1: Partly

Reviewer #2: Yes

2. Has the statistical analysis been performed appropriately and rigorously? 

Reviewer #1: Yes

Reviewer #2: Yes

3. Have the authors made all data underlying the findings in their manuscript fully available?

Reviewer #1: Yes

Reviewer #2: Yes

4. Is the manuscript presented in an intelligible fashion and written in standard English?

Reviewer #1: Yes

Reviewer #2: Yes

5. Review Comments to the Author

Reviewer #1: This study used a sample of consecutive patients (65+) to assess the accuracy of index test in diagnosing pneumonia. The research question is clearly stated. The study design and statistical methods used are appropriate. I recommend publishing this study if the following comments are addressed.

1. The major concern I have is the classification criteria adopted for the usual and reference diagnosis. In both measures, the probability of pneumonia is measured using a three-level Likert scale – low, intermediate, and high. Intermediate and high are then combined to indicate “positive”. The definition of intermediate seems rather vague. It is not clear what “intermediate” actually mean as doctors may have their own subjective interpretations. And since this is then used as the only “gold standard” to construct the ROC curves, the coding scheme may have substantial impacts on the final results. I suggest the authors to report the percentage of intermediate and run a set of sensitivity analysis – that is, whether the AUC values will increase if only “high” is treated as positive case. If they indeed increase considerably, the discussions and conclusions should be revised accordingly. The coding effect needs to be discussed.

2. Although data on a range of clinical characteristics are collected, they are not used in combination with the index tests to improve diagnosis. I think the univariate analysis for sure can provide some important information. But given that the final diagnosis is based on multiple indicators, it is of interest to see whether adding the index tests information in addition to existing criteria can improve the accuracy.

3. Minor comments:

a. In Page 5 Line 7, severity scores of what?

b. In Page 4 Line 15, it is not clear what it means by after completion of the study.

c. Please provide reference for the sample size calculation of the original study. Or add 1-2 sentence to elaborate.

d. Please use the official citation of R:

To cite R in publications use:

R Core Team (2020). R: A language and environment for statistical computing. R Foundation for Statistical Computing, Vienna, Austria. URL https://www.R-project.org/.

e. I think the flow-chart is not necessary as the procedure is quite simple

f. For table 1, please add a column to list the valid sample size for each variable.

Reviewer #2: Summary: This paper examined the ability of serum CRP, PCT, SAA and NP, added to the usual panel of indicators, to improve the diagnostic accuracy of pneumonia in patients over 65 years of age. This is a prospective observational cohort analysis. Of the 899 patients who were screened for inclusion, 200 were eligible based on the presence of one respiratory symptom and one symptom or laboratory value suggestive of infection and a working diagnosis of pneumonia. An expert panel used all available clinical, laboratory and radiographic (all patients had CXR and LDCT) and classified pneumonia as present on 133 patients and absent in 67. A pathogen was identified in 41.4% of patients. The 30-day mortality was 7%. Previous studies suggested CRP and PCT as useful biomarkers for pneumonia in non-elderly population and SAA and NP were more newly suggested but less well studied biomarkers for pneumonia. None were useful in this study to improve diagnosis of pneumonia.

MAJOR CRITICISM: This paper lacks a clearly stated hypothesis. This can be addressed easily in the Introduction using “hypothesized” in place of ‘aimed.” The rationale for examining these biomarkers is presented but could be more concise. These data are well presented and this population is understudied so the results are worthy of publication. The reliance on blood markers to diagnose a process that begins at the lung alveolar surface continues to leave investigators shorthanded. Nonetheless, the authors made a valiant effort and the negative result calls for examination of newer and different biomarkers. Perhaps the authors could comment on the fact that there were more smokers in the pneumonia group and if this could have confounded baseline measures of these biomarkers.

MINOR: English usage could be improved at a few sides in the manuscript.

Example Page 6 line 29: consider “…expert panel classified pneumonia as present in 133 patients and absent in 67….” In place of current sentence.

P5 line 25-6: Consider removal of masculine pronoun and just say “The diagnosis of pneumonia was considered positive if the rated probability was “intermediate” or “high.” Pneumonia was considered negative if it was “low.”

6. PLOS authors have the option to publish the peer review history of their article (what does this mean?). If published, this will include your full peer review and any attached files.

Reviewer #1: No

Reviewer #2: No

---

## [Author Response · Author response to Decision Letter 0]

19 Aug 2020

Dear Editor,

We thank you for the opportunity to submit a revised version of our manuscript. We also thank the reviewers for their thoughtful comments and questions that helped us clarify some points and improve the manuscript. Please, find below our point-by-point answer to the issues raised by the reviewers. Reviewers comments appear in italic and our answers follow in red plain text.

Sincerely yours,

Virginie Prendki on behalf of all authors

Point by point answer to reviewers comments:

Journal Requirements:

Thank you for your help. We have proceeded to the modifications required by PLOS ONE’s style.

2. In your Methods section, please provide additional details regarding the sample size calculation employed to justify the number of patients included in this study.

In the paragraph « Data analysis », we explain that the sample size is based on the power calculation of the original study.

In a previous study [21], a CT scan modified the diagnostic classification of CAP in 59% of cases (95% CI, 53.2–64.0), with an upgraded probability of diagnosis in 19%. Demonstrating an improvement in the pneumonia detection rate by using CT would require 46 patients (p=0.05, power 90%). Considering a true incidence of pneumonia of 45% among patients hospitalised for pneumonia, according to the adjudication committee’s reference diagnosis, we calculated that at least 100 patients would be needed to allow the estimation of any changes in a diagnosis of pneumonia, with a 95% CI.

3. One of the noted authors is a group or consortium: PneumOldCT study group.

In addition to naming the author group, please list the individual authors and affiliations within this group in the acknowledgments section of your manuscript.

Please also indicate clearly a lead author for this group along with a contact email address.

We added the name and affiliations of the authors in the acknowledgments section and the name of the leader (Dr Virginie Prendki, virginie.prendki@hcuge.ch).

5. Review Comments to the Author

Reviewer #1: This study used a sample of consecutive patients (65+) to assess the accuracy of index test in diagnosing pneumonia. The research question is clearly stated. The study design and statistical methods used are appropriate. I recommend publishing this study if the following comments are addressed.

1. The major concern I have is the classification criteria adopted for the usual and reference diagnosis. In both measures, the probability of pneumonia is measured using a three-level Likert scale – low, intermediate, and high. Intermediate and high are then combined to indicate “positive”. The definition of intermediate seems rather vague. It is not clear what “intermediate” actually mean as doctors may have their own subjective interpretations. And since this is then used as the only “gold standard” to construct the ROC curves, the coding scheme may have substantial impacts on the final results. I suggest the authors to report the percentage of intermediate and run a set of sensitivity analysis – that is, whether the AUC values will increase if only “high” is treated as positive case. If they indeed increase considerably, the discussions and conclusions should be revised accordingly. The coding effect needs to be discussed.

We thank the reviewer for this comment. Indeed, the diagnosis of pneumonia is sometimes uncertain, for example when no pathogen is identified and an alternative diagnosis explaining the clinical and radiological data is plausible (eg. Atelectasis, intersitial lung disease, etc)

However, the clinician is finally constrained to make a binary decision: to consider pneumonia as present or not, and to manage the patient accordingly.

As exposed in the Methods section, the assessment of the reference standard was made a posteriori by a panel of experts, using a Delphi method. To reflect the uncertainty surrounding the diagnosis in some patients, the final output was a classification on a 3-levels Likert scale: high, intermediate, or low probability of pneumonia. We then made the assumption that a clinician would often choose to consider a patient with an intermediate probability of disease as having pneumonia and treat him/ her accordingly, as the negative consequences of not treating a pneumonia (false negative) are often considered more severe than treating a patient without pneumonia (false positive). But we agree that this choice is somehow arbitrary.

As proposed, we report the number of patients on the 3-level Likert scale.

“Expert panel classified 99 patients (49.5%) as having high, 34 (17.0%) intermediate, and 67 (33.5%) low probability of disease. Hence, expert panel classified pneumonia as present in 133 patients and absent in 67.” (page 7, lines 14-16)

We made a sensitivity analysis using only the patients in the high probability category as positive for the reference standard. (page 6, lines 11-12)

The results are as follows:

AUROC (95% CI) Ref standard: high or intermediate probability Ref standard: high probability

CRP 0.64 (0.56-0.72) 0.63 (0.56-0.71)

PCT 0.59 (0.51-0.68) 0.61 (0.53-0.69)

SAA 0.60 (0.52-0.69) 0.61 (0.53-0.69)

NP 0.41 (0.32-0.49) 0.45 (0.37-0.54)

CRP/NP 0.63 (0.55-0.71) 0.62 (0.54-0.70)

SAA/ NP 0.61 (0.53-0.70 0.62 (0.54-0.70)

We felt that the AUROC changes observed using the two definitions were not high enough to justify a full report of these results. We added the following sentence in the result section (page 10, lines 15-17)

„The AUROC of the biomarkers using the alternative reference diagnosis definition (considering only the patients with a high probability of disease as positive) were similar and are not reported.“

2. Although data on a range of clinical characteristics are collected, they are not used in combination with the index tests to improve diagnosis. I think the univariate analysis for sure can provide some important information. But given that the final diagnosis is based on multiple indicators, it is of interest to see whether adding the index tests information in addition to existing criteria can improve the accuracy.

Thank you again for helping us to improve the manuscript. Indeed, use of a biomarker should add information on top of commonly collected clinical variables.

Cough (p<0.01), tachypnea (p=0.12), fever (p=0.10), and falls (p=0.10) (as presenting signs or symptoms) were associated with pneumonia with a p value < 0.20. We dismissed falls as it is not widely described as a predictor of pneumonia, and built a score by adding one point for the presence of each of cough, tachypnea and fever. AUROC for the score was 0.63 (95% CI 0.55-0.71). We then dichotomized all biomarkers values at the best predicted cut-off, and computed the AUROC of a new score incorporating the clinical characteristics and each biomarker separately. The results are as follows:

 AUROC clinical score (95% CI) AUROC clinical score + biomarker (95% CI)

CRP (cut-off: 110 mg/L) 0.63 (0.55-0.71) 0.68 (0.60-0.76)

CRP / neopterin (cut-off 15) 0.63 (0.55-0.71) 0.67 (0.59-0.75)

SAA / neopterin (cut-off 37.4) 0.63 (0.55-0.71) 0.66 (0.58-0.74)

SAA (cut-off 282.0 µg/L) 0.63 (0.55-0.71) 0.65 (0.56-0.73)

PCT (cut-off 1.1 µg/L) 0.63 (0.55-0.71) 0.65 (0.57-0.73)

Hence, the addition of each biomarker to clinical variables modestly enhanced the discrimination, a finding already described by van Vugt et al. (BMJ 2013)

However, the absolute value of the AUROC remains low, and it is unlikely that it would alter significantly the diagnostic process for pneumonia.

To address this point, we added the following paragraph in the methods, results, and discussion sections, and a table in the results section:

„To assess if use of any tested biomarker adds information on top of routinely collected clinical variables, we built a clinical score predicting the presence of pneumonia, using clinical symptoms and signs present at admission and associated with the diagnosis in univariate analysis (p< 0.20) and obtained the AUROC of the score with 95% CI. We then added separately each tested biomarker, dichotomized at the best predicted cut-off, to the clinical score, and computed AUROC of the new score“

(page 6, lines 22-27)

Cough, tachypnea, fever, and falls (as presenting signs or symptoms) were associated with pneumonia with a p value < 0.20. We dismissed falls as this is not widely described as a predictor of pneumonia, and built a clinical score by adding one point for the presence of each of cough, tachypnea and fever. AUROC of the clinical score, and AUROCs of scores obtained by adding one point for each dichotomized biomarker to the clinical score are compared in Table 3

Table 3 AUROCS of a clinical score with and without biomarkers for the prediction of pneumonia

 AUROC clinical score (95% CI) AUROC clinical score + biomarker (95% CI)

CRP (cut-off: 110 mg/L) 0.63 (0.55-0.71) 0.68 (0.60-0.76)

CRP / neopterin (cut-off 15) 0.63 (0.55-0.71) 0.67 (0.59-0.75)

SAA / neopterin (cut-off 37.4) 0.63 (0.55-0.71) 0.66 (0.58-0.74)

SAA (cut-off 282.0 µg/L) 0.63 (0.55-0.71) 0.65 (0.56-0.73)

PCT (cut-off 1.1 µg/L) 0.63 (0.55-0.71) 0.65 (0.57-0.73)

(Page 11, lines 8-15)

„Adding a biomarker dichotomized at the best predicted cut-off to a score based on clinical variables (cough, fever and tachypnea) resulted in a higher accuracy. Adding CRP increased AUROC from 0.63 to 0.68 for example. However, the AUROC remains disappointingly low“

(Page 12, lines 9-11)

3. Minor comments:

a. In Page 5 Line 7, severity scores of what?

We added «severity scores of pneumonia”

b. In Page 4 Line 15, it is not clear what it means by after completion of the study.

We modified the sentence for “SAA and NP were measured retrospectively”

c. Please provide reference for the sample size calculation of the original study. Or add 1-2 sentence to elaborate.

This had been added in the paragraph Data Analysis Sample size is based on the power calculation of the original study (reference 5) (page 6 line 15)

d. Please use the official citation of R:

To cite R in publications use:

R Core Team (2020). R: A language and environment for statistical computing. R Foundation for Statistical Computing, Vienna, Austria. URL https://www.R-project.org/.

We have made the change and used the official citation of R (page 7 line 2)

e. I think the flow-chart is not necessary as the procedure is quite simple.

Thank you for this comment. We removed it as proposed.

f. For table 1, please add a column to list the valid sample size for each variable.

This has been done in Table 1.

Reviewer #2: Summary: This paper examined the ability of serum CRP, PCT, SAA and NP, added to the usual panel of indicators, to improve the diagnostic accuracy of pneumonia in patients over 65 years of age. This is a prospective observational cohort analysis. Of the 899 patients who were screened for inclusion, 200 were eligible based on the presence of one respiratory symptom and one symptom or laboratory value suggestive of infection and a working diagnosis of pneumonia. An expert panel used all available clinical, laboratory and radiographic (all patients had CXR and LDCT) and classified pneumonia as present on 133 patients and absent in 67. A pathogen was identified in 41.4% of patients. The 30-day mortality was 7%. Previous studies suggested CRP and PCT as useful biomarkers for pneumonia in non-elderly population and SAA and NP were more newly suggested but less well studied biomarkers for pneumonia. None were useful in this study to improve diagnosis of pneumonia.

MAJOR CRITICISM: This paper lacks a clearly stated hypothesis. This can be addressed easily in the Introduction using “hypothesized” in place of ‘aimed.”

Thank you. We made the proposed change. The end of the Introduction section now reads as follows:

“We hypothesized that biomarkers would improve the clinical and radiological diagnosis of pneumonia, and assessed the accuracy of CRP…” (Page 4, lines 17-18)

The rationale for examining these biomarkers is presented but could be more concise. These data are well presented and this population is understudied so the results are worthy of publication. The reliance on blood markers to diagnose a process that begins at the lung alveolar surface continues to leave investigators shorthanded. Nonetheless, the authors made a valiant effort and the negative result calls for examination of newer and different biomarkers. Perhaps the authors could comment on the fact that there were more smokers in the pneumonia group and if this could have confounded baseline measures of these biomarkers.

We thank the reviewer for his valuable comments.

We commented the fact that there were twice as much smokers in the pneumonia group and added a sentence in the text (page 7 line 10) and in the discussion with a new reference as follows: « Smoking is also associated with alterations in the level of inflammatory markers. In our cohort, smokers were overrepresented in patients with pneumonia.” Ref Shiels et al 2014“

MINOR: English usage could be improved at a few sides in the manuscript.

Example Page 6 line 29: consider “…expert panel classified pneumonia as present in 133 patients and absent in 67….” In place of current sentence.

P5 line 25-6: Consider removal of masculine pronoun and just say “The diagnosis of pneumonia was considered positive if the rated probability was “intermediate” or “high.” Pneumonia was considered negative if it was “low.”

We thank again the reviewer for his help in improving the manuscript. We made the change as asked. We thoroughly reviewed the whole manuscript and changed clumsy formulations at a few sides, without affecting the content.

---

## [Decision Letter · Decision Letter 1]

10 Sep 2020

Accuracy of C-reactive protein, procalcitonin, serum amyloid A and neopterin for low-dose CT-scan confirmed pneumonia in elderly patients: a prospective cohort study

PONE-D-20-15597R1

Dear Dr. Prendki,

We’re pleased to inform you that your manuscript has been judged scientifically suitable for publication and will be formally accepted for publication once it meets all outstanding technical requirements.

Kind regards,

Muhammad Adrish

Academic Editor

PLOS ONE

Additional Editor Comments (optional):

You have satisfactorily addressed all the reviewer's queries.

Reviewers' comments:

Reviewer's Responses to Questions

**Comments to the Author**

1. If the authors have adequately addressed your comments raised in a previous round of review and you feel that this manuscript is now acceptable for publication, you may indicate that here to bypass the “Comments to the Author” section, enter your conflict of interest statement in the “Confidential to Editor” section, and submit your "Accept" recommendation.

Reviewer #1: All comments have been addressed

Reviewer #2: All comments have been addressed

2. Is the manuscript technically sound, and do the data support the conclusions?

Reviewer #1: Yes

Reviewer #2: (No Response)

3. Has the statistical analysis been performed appropriately and rigorously? 

Reviewer #1: Yes

Reviewer #2: (No Response)

4. Have the authors made all data underlying the findings in their manuscript fully available?

Reviewer #1: Yes

Reviewer #2: (No Response)

5. Is the manuscript presented in an intelligible fashion and written in standard English?

Reviewer #1: Yes

Reviewer #2: (No Response)

6. Review Comments to the Author

Reviewer #1: (No Response)

Reviewer #2: (No Response)

7. PLOS authors have the option to publish the peer review history of their article (what does this mean?). If published, this will include your full peer review and any attached files.

Reviewer #1: No

Reviewer #2: No

---

## [Editor Report · Acceptance letter]

18 Sep 2020

PONE-D-20-15597R1

Accuracy of C-reactive protein, procalcitonin, serum amyloid A and neopterin for low-dose CT-scan confirmed pneumonia in elderly patients: a prospective cohort study

Dear Dr. Prendki:

I'm pleased to inform you that your manuscript has been deemed suitable for publication in PLOS ONE. Congratulations! Your manuscript is now with our production department.

Kind regards,

on behalf of

Dr. Muhammad Adrish 

Academic Editor

PLOS ONE